# Design Method Using Response Surface Model for CFRP Corrugated Structure under Quasistatic Crushing

Tetsuya Gomi [1,*], Shotaro Ayuzawa [2], Yuta Urushiyama [2], Kazuhito Misaji [3], Susumu Takahashi [4], Keiichi Motoyama [4,5] and Kosuke Suzuki [4]

[1]  Honda Motor Co., Ltd., 4630 Shimotakanezwawa, Haga-machi, Haga-gun 321-3393, Japan
[2]  Honda R&D Co., Ltd., 4630 Shimotakanezwawa, Haga-machi, Haga-gun 321-3393, Japan; shotaro_ayuzawa@jp.honda (S.A.); yuta_urushiyama@jp.honda (Y.U.)
[3]  College of Industrial Technology Mathematical Information Engineering, Nihon University, 2-1, Izumi-cho 1-chome, Narashino-shi 275-8575, Japan; misaji.kazuhito@nihon-u.ac.jp
[4]  College of Industrial Technology Mechanical Engineering, Nihon University, 2-1, Izumi-cho 1-chome, Narashino-shi 275-8575, Japan; takahashi.susumu32@nihon-u.ac.jp (S.T.); motoyama.keiichi@nihon-u.ac.jp or keiichi@cavs.msstate.edu (K.M.); suzuki.kosuke@nihon-u.ac.jp (K.S.)
[5]  Center for Advanced Vehicular Systems, Mississippi State University, Starkville, MS 39762, USA
*  Correspondence: tetsuya_gomi@jp.honda

**Abstract:** The development of a carbon-fiber-reinforced plastic (CFRP) part is carried out by utilizing many experimental results in deciding the design. For this reason, the development period of a CFRP structure is long and an obstacle for commercialization. In this paper, multiple regression analysis is used to derive a response surface that estimates the generated load using the shape parameters of a corrugated collision energy absorbing structure to shorten the development period. To obtain the response surface, we conducted a quasistatic crushing experiment by using the length of linear portions (pitch) and the number of stacks (thickness) of a corrugated shape as parameters. When progressive crushing mode is observed, energy absorption efficiency decreases with the increase in pitch, and increases with the increase in the number of stacks. To discuss how energy absorption efficiency changes, a comparison examination is conducted using the derived response surfaces. Results indicate that specifications with high energy absorption efficiency can be accurately selected using the response surface of primary expression. In addition, differences in deformation mode were due to the influence of the stress at the corner portion of a part.

**Keywords:** CFRP; energy absorption; progressive crushing; compression; corrugate structure; response surface; design method

## 1. Introduction

To address global environmental problems, continuous efforts are being made for $CO_2$ reduction. In the transport sector, which accounts for approximately 14% of $CO_2$ emissions, electrification and aerodynamic drag reduction and rolling resistance reduction have been promoted to meet the demand for reduced $CO_2$ emissions from driving vehicles [1]. For electric vehicles, which will be more widely applied, weight reduction in the vehicle body is considered to compensate for the increased weight of parts needed for electrification, and to reduce rolling resistance. Since collision performance is also important, structures that take account of weight reduction and collision safety performance are studied [2,3].

For weight reduction that considers collision safety performance, strengthening high-tensile-strength steel, and the application of aluminum alloys and fiber-reinforced plastic (FRP) have attracted attention [4]. For metal materials, such as steel and aluminum alloy sheets, structures that use plastic deformation such as buckling and bending are generally used to increase collision energy absorption. Since FRP utilizes the generated energy during fiber breakage, delamination, and other breaking, it shows excellent energy

absorption performance. Structures, particularly cylindrical, have a destruction mode called "progressive crushing" in which breaking continuously progresses with the inner and outer sides of a wall face rolling back in mutually opposite directions [5,6]. Figure 1 shows perspective and enlarged views of the end portion of a structure in representative progressive crushing destruction mode.

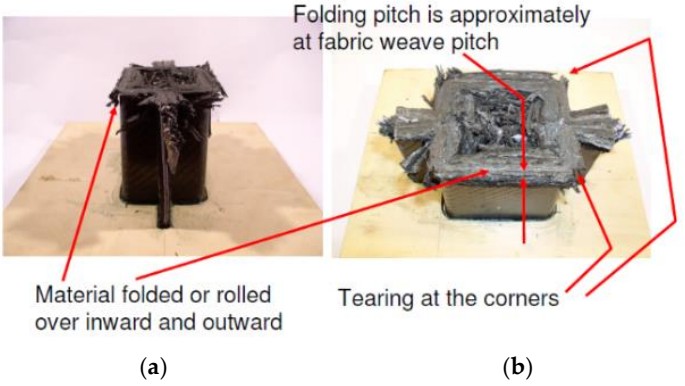

(**a**)                                           (**b**)

**Figure 1.** Progressive crushing destruction mode of FRP structure. (**a**) Perspective view; (**b**) enlarged view of the end.

In this destruction mode, as the breaking of fiber and resin continuously progresses, it has the characteristic of the load–displacement curve used for the calculation of energy absorption having an ideal rectangular shape. Figure 2 shows a typical example of a load–displacement curve obtained when progressive crushing destruction mode is observed. For displacement, the load end position was set to 100%.

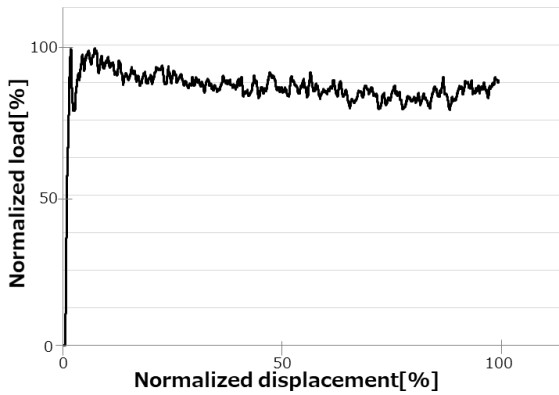

**Figure 2.** Load–displacement curve of FRP structure (typical).

Many studies on an energy absorbing structure that uses FRP in automotive structures tend to assume frontal or rear-end collision and are related to cylindrical or hat-shaped parts [7]. These studies have significantly deepened the understanding about collision phenomena of part of an existing structure when its material is changed. However, understanding the phenomena alone is not sufficient when developing an FRP structure. In actual development, performance is predicted by utilizing the results of many experiments, including coupon tests and component tests [8]. Because of this, when there is a design change, it is sometimes necessary to conduct additional experiments of the CFRP structure. On this basis, the development period of a CFRP structure is long and an obstacle for commercialization.

Expectations are running high that FRP can reduce vehicle body weight. One example is a structure that protects the battery from a side collision. To test this structure, the test conditions of FMVSS no. 214 Rigid Pole Side Impact Test are used, in which the side face of

a vehicle collides with a solid pole barrier 10 inches in diameter at the speed of 20 mph [9]. Figure 3 shows the positions of the solid pole barrier and vehicle during the test.

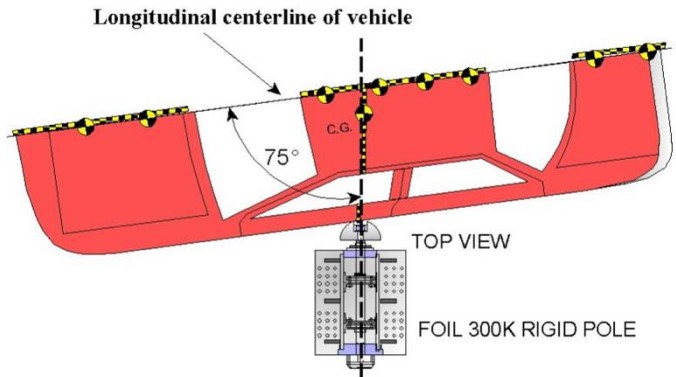

**Figure 3.** Pole barrier set position in impact test [9]. Available online: https://one.nhtsa.gov/staticfiles/nvs/pdf/TP214P-01_APP_B_CHECKSHEETS.pdf (accessed on 25 October 2021).

Figure 4 shows the result of a rigid pole impact test of an internal combustion engine vehicle (ICEV) [10].

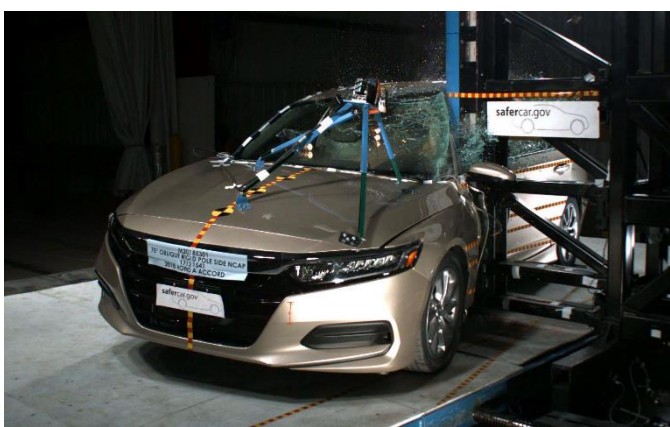

**Figure 4.** Dynamic maximal deformation in impact test [10]. Available online: https://www.nhtsa.gov/vehicle/2018/HONDA/ACCORD/4%252520DR/FWD (accessed on 25 October 2021).

In the case of battery electric vehicles, a battery box is often positioned under the floor. It is desirable to further suppress the floor deformation during the impact test than in an ICEV to increase battery mounting capacity. Deformation is generally suppressed by increasing the thickness of the steel parts used for the vehicle body to increase the generated load at the time of collision to increase the amount of absorbed energy. This generally used approach, however, results in an increase in vehicle body weight. Given the situation, a corrugated collision energy absorbing part composed of CFRP that can be used as a structure that absorbs energy from a side collision was selected as the subject for this study. The mechanical behavior of CFRP structures is severely affected by the loading rate due to the influence of polymer-based materials [11–13]. Considering the collision form in the real world, static performance and impact performance are taken into consideration in collision energy absorption structure design [14–17]. This paper proposes a design technology for quasistatic performance. Figure 5 shows the installation location of the corrugated structural part for this study. The corrugated structural member is placed in the energy-absorbing (EA) area that absorbs energy from the time when the door panel comes into contact with the solid pole barrier to the time when the vehicle stops.

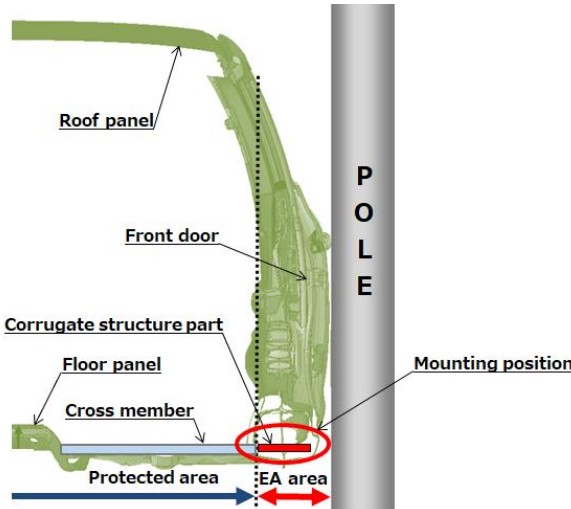

**Figure 5.** Layout of support structure of vehicle body.

After obtaining a response surface by clarifying the relation of the shape parameters to the absorbed energy and generated load, we propose a method for deciding the dimensions of a corrugated energy absorbing structure while satisfying the required conditions by using the derived response surface.

## 2. Methods

### 2.1. Material and Test Specimen

We used a carbon fiber prepreg cloth TR3523 (Epoxy) made by Mitsubishi Chemical Corporation as the material of the corrugated structural part.

Figure 6 shows the cross-sections of the specimens. We prepared specimens in the following four patterns by changing the pitch along the horizontal surface length of the cross-section: 5 mm (P5), 10 mm (P10), 15 mm (P15), and 20 mm (P20). Cross-sectional height was set to be 17 mm, and draft angle 15 degrees in consideration of the application location of the part for this study, as shown in Figure 5. Thickness was established by stacking the carbon fiber prepreg. Three levels of thickness, namely, 8-, 12-, and 16-ply, were used.

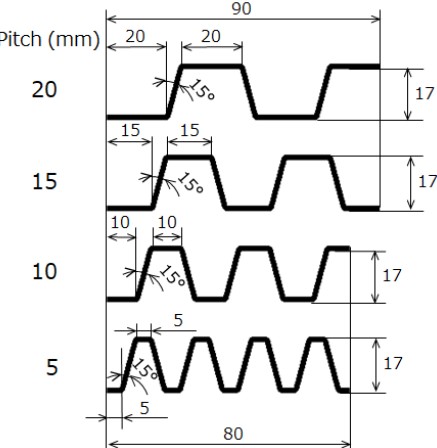

**Figure 6.** Cross-section of corrugate structure.

Figure 7 shows a perspective view with the respective dimensions. Stacks were configured to have a 0/90 orientation with the axial direction of 0 degrees, and orthogonal of 90 degrees. Specimens were formed with an autoclave.

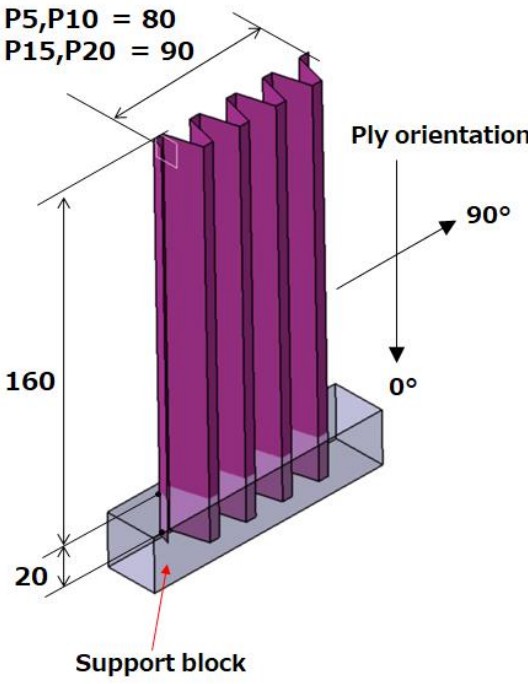

**Figure 7.** Dimensions of corrugate structure.

Figure 8 shows a specimen that was adhesion-fixed onto a support block. One end of the specimen was machined into an angle of 45 degrees to serve as a triggering portion. This triggering portion is effective in inducing early breaking [18]. The portion of the specimen to be bonded was embedded into an aluminum alloy pedestal to the depth of 20 mm and then adhesively fixed with epoxy resin.

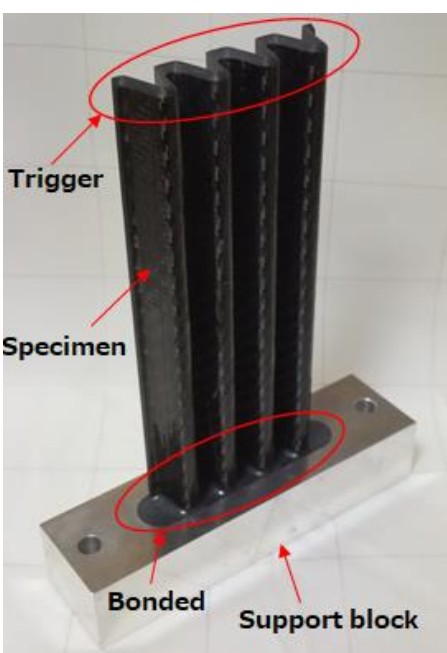

**Figure 8.** Corrugate structure with support block.

Table 1 shows mechanical properties of the carbon fiber prepreg cloth TR3523 (Epoxy) used.

**Table 1.** Mechanical properties of CFRP laminate [0/90].

| Mechanical Property | | Values |
|---|---|---|
| Density [g/cm$^3$] | | 1.50 |
| Compression (SACMA SRM6) | Young's modulus (GPa) | 59.49 |
| | Ultimate strength (Mpa) | 671.20 |
| | Poisson's ratio | 0.04 |
| Tension (ASTM D 3039) | Young's modulus (GPa) | 62.47 |
| | Ultimate strength (Mpa) | 869.57 |
| | Poisson's ratio | 0.04 |

Table 2 shows the design values of the specimens.

**Table 2.** Structural parameters.

| Number of Stacks (ply) | Lay-Up | Thickness (mm) | Pitch (mm) |
|---|---|---|---|
| 8 | [0/90]$_8$ | 1.82 | 5, 10, 15, 20 |
| 12 | [0/90]$_{12}$ | 2.72 | Same |
| 16 | [0/90]$_{16}$ | 3.63 | Same |

*2.2. Quasistatic Compression Test*

A large 600 kN universal test system of Instron 5589 series was used as the compression test equipment. A planar shape was selected as the surface shape of the loading plate. Compaction speed was set at 50 mm/min. The test displacement was set to be 140 mm from the contact point between the loading plate and the specimen. Figure 9 shows a specimen mounted onto the test equipment.

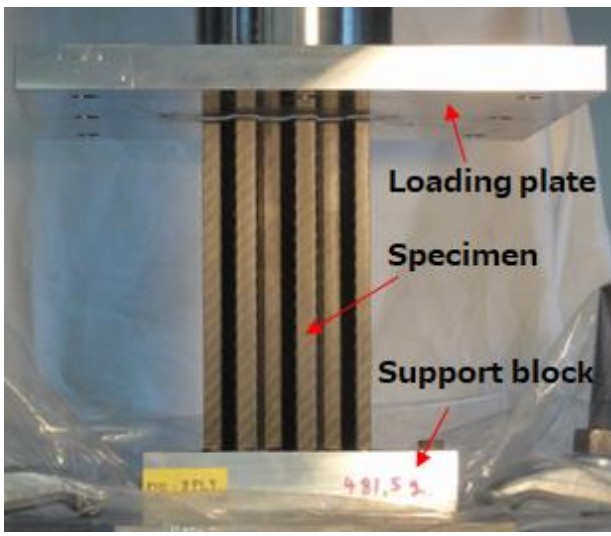

**Figure 9.** Experimental setup for quasistatic compression test.

### 3. Test Results

*Specimen Load Characteristics and Observed Results*

The obtained load–displacement curves through the compression test are as shown below. Figures 10–12 show the load–displacement curves of each pitch in the 8-, 12-, and 16-ply specimens, respectively.

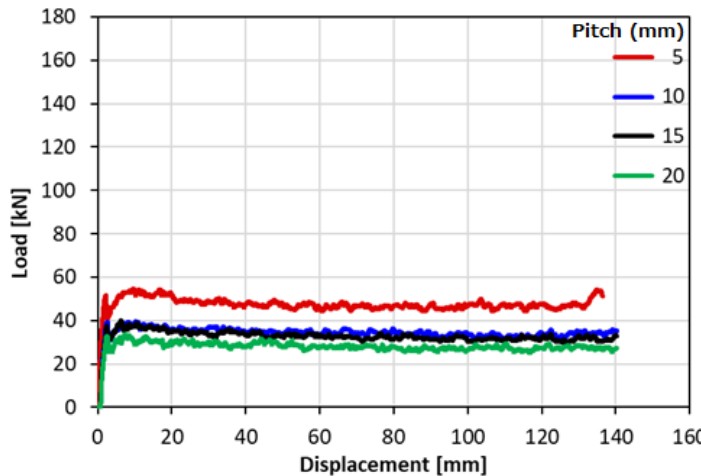

**Figure 10.** Load–displacement curves of 8 ply.

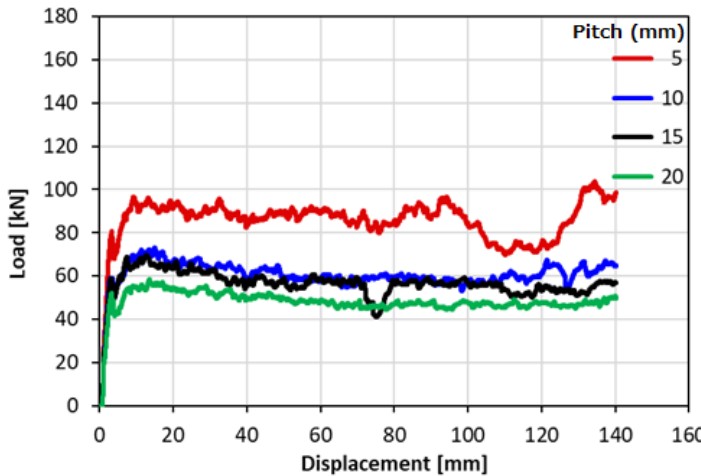

**Figure 11.** Load–displacement curves of 12 ply.

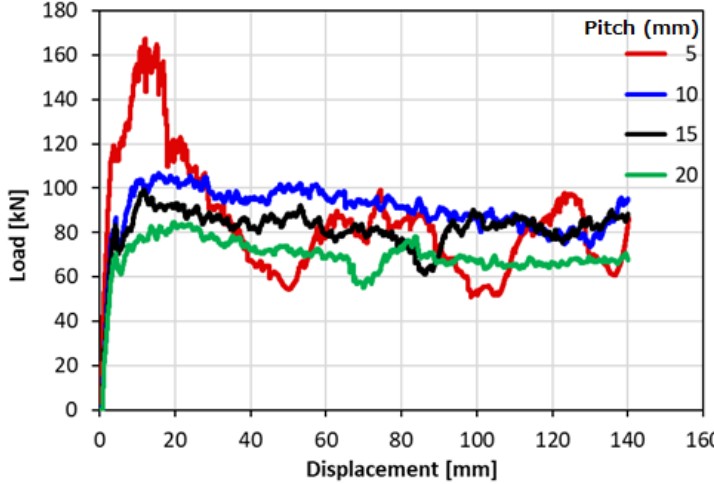

**Figure 12.** Load–displacement curves of 16 ply.

In this test, progressive crushing mode was observed in all conditions except for 16-ply with pitch of 5 mm (16ply-P5). As shown in Figure 12, in the load–displacement curve of 16ply-P5, the generated load was high in the early stage, but subsequently dropped. Comparisons were conducted using the recorded images of the compression test process

to confirm differences in deformation mode. Figure 13 shows the deformation process of 12ply-P15 in progressive crushing mode, and Figure 14 shows the deformation process of 16ply-P5, which was not in progressive crushing mode.

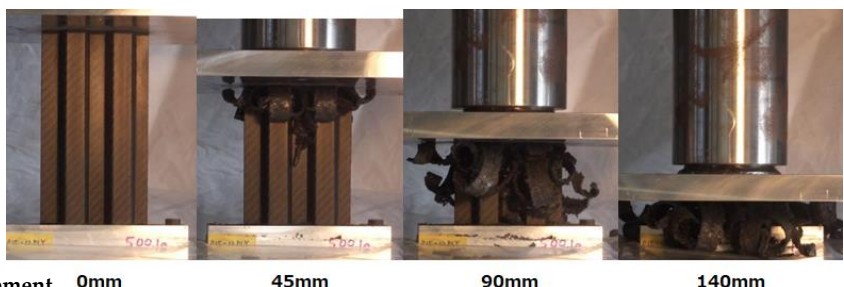

**Figure 13.** Progressive crushing procedures of compression test (case: 12ply-P15).

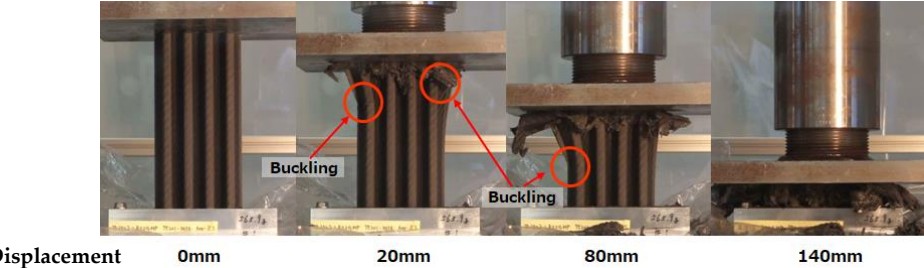

**Figure 14.** Nonprogressive crushing procedures of compression test (case: 16ply-P5).

The 12ply-P15 specimen continuously crushed in progressive crushing mode from the portion that came into contact with the loading plate. The 16ply-p5 specimen, which did not crush in progressive crushing mode, crushed intermittently, with the ridge portions becoming buckled and crushed. Figure 15 shows the specimens after the test.

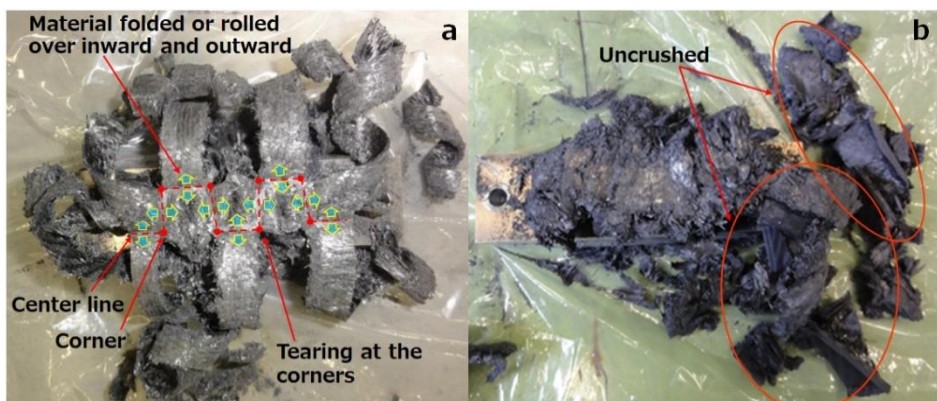

**Figure 15.** Comparison of specimen after compression test. (**a**) Progressive crushing, case: 12ply-P15; (**b**) nonprogressive crushing, case: 16ply-P5.

## 4. Analysis and Discussion

### 4.1. Mathematical Model of Energy Absorption Efficiency

Energy absorption (EA) is the total energy obtained from a load–displacement curve. As the evaluation zone of EA, we selected the displacement range between 0 and 60 mm, where the load had been generated in the experiment. EA is represented by Expression (1), where F is generated load, and s is displacement.

$$\mathrm{EA} = \int_0^{60} F \cdot ds \qquad (1)$$

Energy absorption efficiency (EAE) is defined as the obtained value by dividing EA until 60 mm of the specimen becomes deformed by mass m of the deformed 60 mm portion of a corrugated structure pert. This is represented by Expression (2). The value of EAE is large when efficiency is high.

$$\text{EAE} = \frac{EA}{m} \tag{2}$$

Figure 16 shows the relation between EAE and the pitch with the varied numbers of stacks. As shown in Figure 16, EAE mostly linearly decreased with the increase in pitch in each number of stacks, except for 16ply-P5. EAE increased with the increase in the number of stacks.

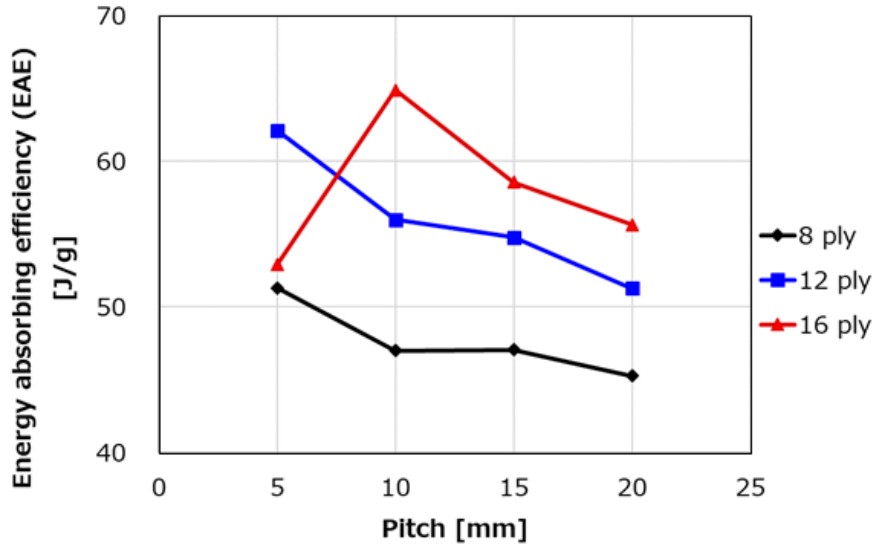

**Figure 16.** EAE–pitch curves.

Multiple regression analysis was conducted, with the number of stacks and pitch serving as explanatory variables and EAE as a response variable. Table 3 shows the explanatory variables and response variable.

**Table 3.** Explanatory variable and response variable (EAE).

| Explanatory Variable | | Response Variable |
| --- | --- | --- |
| Number of Stacks (ply) | Pitch (mm) | EAE (J/g) |
| | 5 | 51.3 |
| 8 | 10 | 47.0 |
| | 15 | 47.1 |
| | 20 | 45.3 |
| | 5 | 62.1 |
| 12 | 10 | 56.0 |
| | 15 | 54.8 |
| | 20 | 51.3 |

**Table 3.** *Cont.*

| Explanatory Variable | | Response Variable |
|---|---|---|
| Number of Stacks (ply) | Pitch (mm) | EAE (J/g) |
| | 5 | 52.9 |
| | 10 | 64.9 |
| 16 | 15 | 58.5 |
| | 20 | 55.6 |

Multiple regression analysis was conducted on all experiment results and only on those that resulted in progressive crushing mode. For EAE, a linear model was used because the relation between the number of stacks and pitch mostly linearly varied. The model expression is represented by Expression (3).

$$EAE = \beta_0 + \beta_1 S + \beta_2 P \qquad (3)$$

where S is the number of stacks, P is pitch, $\beta_0$ is intercept, $\beta_1$ is partial regression coefficient of the number of stacks, and $\beta_2$ is partial regression coefficient of the pitch.

Table 4 shows the multiple regression analysis results conducted on all experiment results. Results were considered to be statistically significant when the *p* value was below 0.05.

**Table 4.** Multiple regression analysis results using all results (EAE).

| Item | Partial Regression Coefficient ($\beta_0$, $\beta_1$, $\beta_2$) | *p* Value | Std. Error | 95% Confidence Interval | |
|---|---|---|---|---|---|
| | | | | Lower | Upper |
| Intercept | 42.6 *** | *p* < 0.001 | 5.14 | 31.0 | 54.2 |
| Number of stacks | 1.29 ** | 0.00564 | 0.356 | 0.481 | 2.09 |
| Pitch | −0.332 | 0.145 | 0.208 | −0.803 | 0.139 |
| Multiple R-squared ($R^2$) | | | 0.634 | | |
| Adjusted R-squared ($R'^2$) | | | 0.553 | | |

*** Coefficient estimated with statistical significance of 0.001; ** coefficient estimated with statistical significance of 0.01.

In this model, since the *p* value of the pitch was larger than 0.05, and 0 was included in the 95% confidence interval, the response variable was not explained by the explanatory variables. To improve prediction accuracy, multiple regression analysis was conducted by excluding the experimental results of 16ply-P5, which had not resulted in progressive crushing mode. Table 5 shows the multiple regression analysis results excluding 16ply-P5 experimental results.

**Table 5.** Multiple regression analysis results using progressive crushing mode results (EAE).

| Item | Partial Regression Coefficient ($\beta_0$, $\beta_1$, $\beta_2$) | *p* Value | Std. Error | 95% Confidence Interval | |
|---|---|---|---|---|---|
| | | | | Lower | Upper |
| Intercept | 42.0 *** | *p* < 0.001 | 2.47 | 36.3 | 47.7 |
| Number of stacks | 1.71 *** | *p* < 0.001 | 0.187 | 1.28 | 2.14 |
| Pitch | −0.601 *** | *p* < 0.001 | 0.111 | −0.857 | −0.345 |

**Table 5.** *Cont.*

| Item | Partial Regression Coefficient ($\beta_0$, $\beta_1$, $\beta_2$) | *p* Value | Std. Error | 95% Confidence Interval | |
|---|---|---|---|---|---|
| | | | | Lower | Upper |
| Multiple R-squared ($R^2$) | | | 0.925 | | |
| Adjusted R-squared ($R'^2$) | | | 0.906 | | |

*** Coefficient estimated with a statistical significance of 0.001.

This model indicates good values with determination coefficient $R^2$ being 0.925, and the adjusted coefficient of determination $R'^2$ being 0.906. In addition, since the *p* values of this model were smaller than 0.05, and 0 was not included in the 95% confidence interval, the response variable was explained by the explanatory variables. This model, therefore, represents the response surface to be used to obtain EAE. Figure 17 shows the response surface obtained using the estimation equation and the test results. The red asterisk shown in the figure represents the excluded results of 16ply-P5.

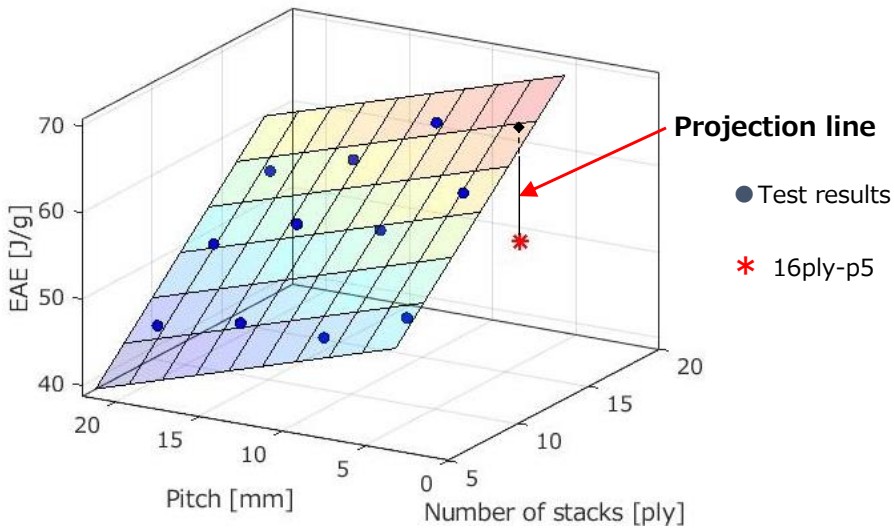

**Figure 17.** Response surface and test results (EAE).

### 4.2. Mathematical Model of Crushing Force per Unit Length

In developing the part that is this study's subject, the targeted energy absorption and displacement were determined, and these define the generated load and decide the specifications of the parts. It is, therefore, necessary to derive a response surface to be used to obtain the generated load. In obtaining the crushing force per unit length (CFL), the mean crushing force (MCF) of a specimen is used. We used the average crushing force in the displacement range between 10 and 60 mm as the MCF. CFL is obtained by dividing MCF by the cross-sectional circumferential length of the corrugated shape. CFL is represented by Expression (4), where L is the cross-sectional circumferential length.

$$\text{CFL} = \frac{MCF}{L} \tag{4}$$

Multiple regression analysis was conducted with the number of stacks and pitch as explanatory variables, and CFL as a response variable.

Table 6 shows the explanatory variables and response variable.

**Table 6.** Explanatory variable and response variable (CFL).

| Explanatory Variable | | Response Variable |
|---|---|---|
| **Number of Stacks (ply)** | **Pitch (mm)** | **CFL (kN/mm)** |
| | 5 | 0.275 |
| | 10 | 0.255 |
| 8 | 15 | 0.249 |
| | 20 | 0.234 |
| | 5 | 0.506 |
| | 10 | 0.456 |
| 12 | 15 | 0.437 |
| | 20 | 0.412 |
| | 5 | 0.527 |
| | 10 | 0.710 |
| 16 | 15 | 0.635 |
| | 20 | 0.605 |

Multiple regression analysis was conducted on both of the following patterns: all experimental results, and only the results in progressive crushing mode. On the basis of Expression (1), CFL, which was proportionate to F, was proportionate to EAE, which was proportionate to EA. Since a linear model was used for EAE because the relation between the number of stacks and the pitch mostly linearly varied, a linear model was also used for CFL. Expression (5) shows the model expression for obtaining CFL.

$$CFL = \beta_0 + \beta_1 S + \beta_2 P \tag{5}$$

where S is the number of stacks, P is pitch, $\beta_0$ is intercept, $\beta_1$ is partial regression coefficient of the number of stacks, and $\beta_2$ is partial regression coefficient of the pitch.

Table 7 shows the multiple regression analysis results conducted by using all experimenta; results.

**Table 7.** Multiple regression analysis results using all results (CFL).

| Item | Partial Regression Coefficient ($\beta_0$, $\beta_1$, $\beta_2$) | *p* Value | Std. Error | 95% Confidence Interval | |
|---|---|---|---|---|---|
| | | | | **Lower** | **Upper** |
| Intercept | −0.0847 | 0.215 | 0.0635 | −0.228 | 0.0589 |
| Number of stacks | 0.0458 *** | *p* < 0.001 | 0.00440 | 0.0358 | 0.0557 |
| Pitch | −0.00181 | 0.500 | 0.00257 | −0.00762 | 0.00401 |
| Multiple R-squared ($R^2$) | | | 0.923 | | |
| Adjusted R-squared ($R'^2$) | | | 0.906 | | |

*** Coefficient estimated with statistical significance of 0.001.

In this model, since the *p* values of the pitch and intercept were larger than 0.05, and 0 was included in the 95% confidence interval, the response variable was not explained by the explanatory variables. Subsequently, multiple regression analysis was conducted by excluding the experimental results of 16ply-P5 that had not resulted in progressive crushing mode. Table 8 shows the multiple regression analysis results conducted by excluding the experimental results of 16ply-P5.

**Table 8.** Multiple regression analysis results using progressive crushing mode results (CFL).

| Item | Partial Regression Coefficient ($\beta_0$, $\beta_1$, $\beta_2$) | p Value | Std. Error | 95% Confidence Interval | |
|---|---|---|---|---|---|
| | | | | Lower | Upper |
| Intercept | −0.0919 ** | 0.00594 | 0.0248 | −0.149 | −0.0348 |
| Number of stacks | 0.0512 *** | $p < 0.001$ | 0.00188 | 0.0468 | 0.0555 |
| Pitch | −0.00528 ** | 0.00146 | 0.00111 | −0.00784 | −0.00271 |
| Multiple R-squared ($R^2$) | | | 0.989 | | |
| Adjusted R-squared ($R'^2$) | | | 0.987 | | |

*** Coefficient estimated with statistical significance of 0.001; ** coefficient estimated with statistical significance of 0.01.

This model indicates good values with determination coefficient $R^2$ being 0.989, and adjusted coefficient of determination $R'^2$ being 0.987. In addition, since the *p* values of this model were smaller than 0.05, and 0 was not included in the 95% confidence interval, the response variable was explained by the explanatory variables. This model, therefore, represents a response surface to be used to obtain CFL. Figure 18 shows the response surface obtained using the estimation equation and the test results. The red asterisk shown in the figure represents the excluded results of 16ply-P5.

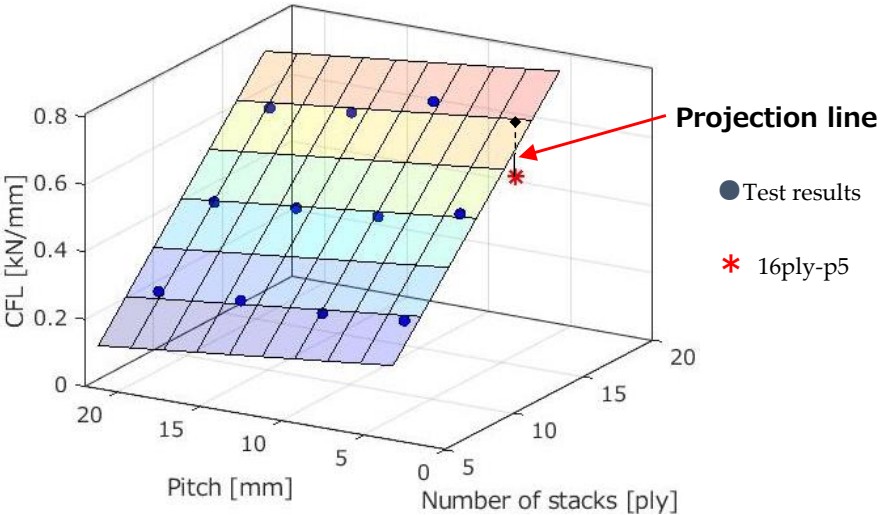

**Figure 18.** Response surface and test results (CFL).

The area within the red frame in Figure 19 is defined as the range that could be used for the design.

### 4.3. Understanding the Phenomena Using the Stress Model of Corner Portions and Linear Portions

To understand the phenomena, multiple regression analysis was conducted by changing the explanatory variables. The cross-sectional areas of the corner and linear portions could be obtained by replacing the explanatory variables of pitch and of the number of stacks with the geometric elements of corrugated cross-section. As shown in Figure 20, the cross-sectional area of a corner portion is defined as the area between the lines drawn by connecting the inner and outer side ends of the radius. A partial regression coefficient, obtained by conducting multiple regression analysis without intercept and with the use of cross-sectional area of the corner portion and cross-sectional area of the linear portion as explanatory variables and MCF as the response variable, represents the stress.

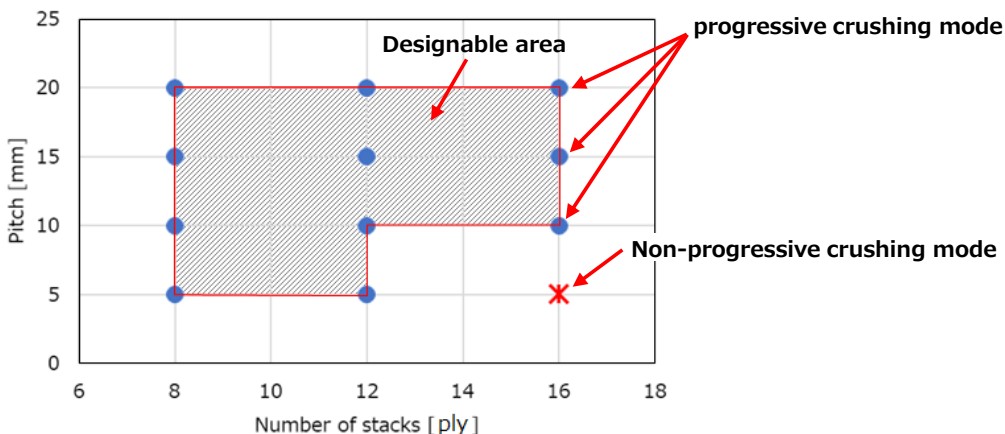

**Figure 19.** Designable area of corrugate structure.

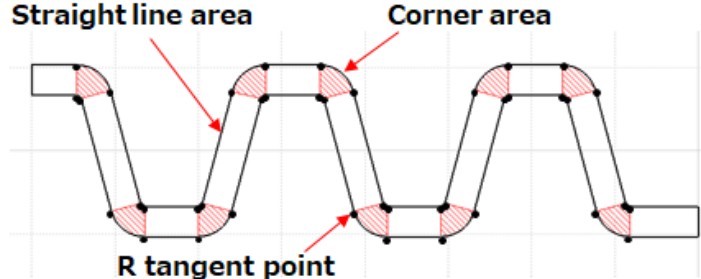

**Figure 20.** Straight-line and corner areas of corrugate structure.

Table 9 shows the explanatory variables and response variable.

**Table 9.** Explanatory variable and response variable (MCF).

| Case | | Explanatory Variable | | Response Variable |
|---|---|---|---|---|
| Number of Stacks (ply) | Pitch (mm) | Corner Area (CA) (mm$^2$) | Straight-Line Area (LA) (mm$^2$) | MCF (kN) |
| 8 | 5 | 72.4 | 242 | 48.8 |
| | 10 | 45.2 | 204 | 35.7 |
| | 15 | 36.2 | 209 | 34.4 |
| | 20 | 27.1 | 196 | 29.3 |
| 12 | 5 | 108.0 | 363 | 89.6 |
| | 10 | 67.8 | 306 | 63.8 |
| | 15 | 54.3 | 313 | 60.2 |
| | 20 | 40.7 | 294 | 51.8 |
| 16 | 10 | 90.5 | 408 | 99.4 |
| | 15 | 72.4 | 418 | 87.5 |
| | 20 | 54.3 | 393 | 75.9 |

On the basis of Expression (1), MCF, which was proportionate to F, was proportionate to EAE, which was proportionate to EA. Since a linear model was used for EAE because the relation between the number of stacks and the pitch mostly linearly varied, a linear model was also used for MCF. Expression (6) shows the model expression for obtaining MCF.

$$\text{MCF} = \beta_1 \text{CA} + \beta_2 \text{LA} \tag{6}$$

where CA is the corner portion area, LA is the linear portion area, $\beta_1$ is the partial regression coefficient of the corner portion area, and $\beta_2$ is the partial regression coefficient of the linear portion area.

Table 10 shows the multiple regression analysis results conducted by using all experiment results.

**Table 10.** Multiple regression analysis results using all results (MCF).

| Item | Partial Regression Coefficient ($\beta_1$, $\beta_2$) | *p* Value | Std. Error | 95% Confidence Interval | |
|---|---|---|---|---|---|
| | | | | Lower | Upper |
| Corner area (CA) | 0.163 | 0.196 | 0.118 | −0.0989 | 0.425 |
| Straight-line area (LA) | 0.171 *** | $p < 0.001$ | 0.0269 | 0.111 | 0.231 |
| Multiple R-squared ($R^2$) | | | 0.987 | | |
| Adjusted R-squared ($R'^2$) | | | 0.985 | | |

*** Coefficient estimated with statistical significance of 0.001.

In this model, the partial regression coefficient of LA was good, but the *p* value of CA was larger than 0.05, and 0 was included in the 95% confidence interval. The response variable was, therefore, not explained by the explanatory variables.

Subsequently, multiple regression analysis was conducted by excluding the experiment results of 16ply-P5 that did not result in progressive crushing mode. Table 11 shows the multiple regression analysis results conducted by excluding 16ply-P5 experiment results.

**Table 11.** Multiple regression analysis results using progressive crushing mode results. (MCF).

| Item | Partial Regression Coefficient ($\beta_1$, $\beta_2$) | *p* value | Std. Error | 95% Confidence Interval | |
|---|---|---|---|---|---|
| | | | | Lower | Upper |
| Corner area (CA) | 0.332 * | 0.0234 | 0.122 | 0.0565 | 0.608 |
| Straight-line area (LA) | 0.142 *** | $p < 0.001$ | 0.0256 | 0.0843 | 0.200 |
| Multiple R-squared ($R^2$) | | | 0.991 | | |
| Adjusted R-squared ($R'^2$) | | | 0.989 | | |

*** Coefficient estimated with a statistical significance of 0.001, * coefficient estimated with a statistical significance of 0.05.

Since the *p* values of this model were smaller than 0.05, and 0 was not included in the values in the 95% confidence interval, the response variable was explained by the explanatory variables.

These two models were compared to discuss the phenomena in which the deformation model is different. The stress of the linear portions indicated by the partial regression coefficient of LA was considered to be statistically significant, whether or not progressive crushing mode was observed. The stress was obtained by multiplying the partial regression coefficient by 1000 because the unit of response variable MCF was kN. The predicted stress on the basis of the two models was 171 and 142 MPa. However, as to the partial regression coefficient of CA obtained from the results of all experiments, the exclusion of data not in progressive crushing mode would improve the *p* value to achieve a level that is statistically significant. The stress of this model was 332 MPa, and this was 2.34 times higher than the stress of the linear portions of 142 MPa. On the above basis, the stress at the corner portions largely contributed in deciding whether collapse occurs in progressive crushing mode.

In recent years, technical studies have been conducted using computer-aided engineering (CAE) to facilitate the development of FRP parts. In these studies, prediction accuracy is verified by recreating the experiments with CAE and comparing the generated

load and deformation mode [19,20]. Since the results obtained this time could be used as a new confirmation item in verifying prediction accuracy, they are helpful in enhancing CAE technology.

## 5. Conclusions

In a corrugated energy absorbing specimen that is formed with CFRP to be subject to static compression, we examined a development method in which specifications are determined with a response surface.

On the above basis, experiments showed that EAE increases when the pitch is reduced, and the number of stacks is increased. Furthermore, we proposed a method to develop a corrugated part using a response surface and verified the effectiveness by showing that EAE can be represented by a linear model when varying the pitch and the number of stacks. In the region of a response surface defined with the explanatory variables of pitch and the number of stacks that can be used for the design, it is unnecessary to conduct additional experiments for the understanding of characteristics of a CFRP structure part. This demonstrates that the development time and cost of corrugated energy-absorbing parts can be reduced.

The obtained insights are as summarized below.

1. In progressive crushing mode, EAE decreases with the increase in pitch.
2. In progressive crushing mode, EAE increases with the increase in the number of stacks.
3. In nonprogressive crushing mode, EAE significantly decreases.
4. The energy absorption efficiency and average load generated in specimens with the varying number of stacks and pitch can be accurately represented by a linear response surface.
5. As a result of examination by noting whether or not progressive crushing mode is observed, the corner portions largely contributed, with the predicted stress being 2.34 times larger than that of the linear portions.

**Author Contributions:** Conceptualization, S.A. and Y.U.; methodology, S.A.; validation, S.A.; writing-review and editing, T.G., Y.U., K.M. (Kazuhito Misaji), S.T., K.M. (Keiichi Motoyama) and K.S.; supervision, K.M. (Kazuhito Misaji). All authors have read and agreed to the published version of the manuscript.

**Funding:** This research received no external funding.

**Institutional Review Board Statement:** Not applicable.

**Informed Consent Statement:** Not applicable.

**Data Availability Statement:** Not applicable.

**Conflicts of Interest:** The authors declare no conflict of interest.

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
