# Peer review of "Design Method Using Response Surface Model for CFRP Corrugated Structure under Quasistatic Crushing"

_applsci, doi:10.3390/app112110178_

Round 1

Reviewer 1 Report

"Design method using response surface model for CFRP corrugated structure under quasi-static crushing" is an interesting and well-written study regarding the progressive crushing of corrugated CFRP structures for use in vehicles energy-absorbing elements. An experimentally derived model is proposed to decrease the number of experiments necessary.

However - and this is the most important drawback of the study - no consideration is given to the effects of loading rate on the properties of polymer-based materials. Polymers, that constitute the matrix of CFRP, are viscoelastic materials whose mechanical behavior is greatly affected by the loading rate, as is demonstrated time and time again by various studies. The composites in the study were tested at the rate of 50 mm/min, while an impact test is shown with a velocity of 20 mph - this is different by 4 orders of magnitude.  Therefore, the applicability of the study directly to impact-absorbing structures is doubtful. Do impacted structures exhibit the same linear behavior? We do not know and we will not learn it from the study.

More specific comments:

1 - The literature included in the paper is extremely limited. There exist a vast body of literature regarding the progressive crushing of FRP materials, almost none of which is used here. A more thorough literature review is advisable.

2 - Figure 2 uses normalized displacement [%] on a horizontal axis. Does 100% displacement mean the total length of the specimen was crushed without any increase of the force at the end?

3 - More info on the prepreg is needed. What is the reinforcement (uni-directional fabric? biaxial? multiaxial?)? What is the resin?

4 - More info on ply orientation is needed. Where all of them oriented in the same direction?

5 - Is only 1 specimen used for every pitch and ply number? Can we be sure how repeatable are the results?

6 - Can the Authors propose a mechanism for why does the 16ply-P5 behaves differently from other specimens?

7 - Tables 4, 5, 7, 8, 10, and 11, as well as the text surrounding them, reference p values which are not shown in the tables. It would be better if the tables included the actual p values instead (or beside) the asterisk notation about the significance of the results.

8 - Figures 17 and 18 include the response surface, as well as the asterisk representing the outlying point. However, it is impossible to tell how does the asterisk corresponds to the surface in 3-dimensional space - does it lie above, below, or beside the plane?

Author Response

Dear Reviewer,

Thank you for providing these insights.
I think that your point made it possible to make the treatise easy to understand.
The response to comments is specified below. The corrections in the text are shown in red.

>no consideration is given to the effects of loading rate on the properties of >polymer-based materials.

As you said, there was no mention of polymer-based materials mechanical behavior being affected by loading rate.

In the design of automobile parts, the collision form in the real world is considered, and in the collision energy absorption structure design, static performance and impact performance are considered. Therefore, this paper proposes a design technique for quasi-static performance. We added papers such as research on the static dynamic ratio of resin and crash boxes to the references.(line90-line94)

We would like to discuss the verification of dynamic performance and the explanation of the mechanism in the next paper.

>Figure 2 uses normalized displacement [%] on a horizontal axis. Does 100% >displacement mean the total length of the specimen was crushed without any >increase of the force at the end?

 For displacement, the load end position was set to 100%.(Described on line 60.)

>More info on the prepreg is needed. What is the reinforcement (uni-directional >fabric? biaxial? multiaxial?)? What is the resin? More info on ply orientation is >needed. Where all of them oriented in the same direction?

[0/90] Cloth material and epoxy resin are used. Added resin name and laminating information to the text.(Described on line 60 and Table 2.)

>Tables 4, 5, 7, 8, 10, and 11, as well as the text surrounding them, reference p >values which are not shown in the tables. It would be better if the tables >included the actual p values instead (or beside) the asterisk notation about the >significance of the results.

Added p-value to Tables 4, 5, 7, 8, 10, and 11.

>Figures 17 and 18 include the response surface, as well as the asterisk >representing the outlying point. However, it is impossible to tell how does the >asterisk corresponds to the surface in 3-dimensional space - does it lie above, >below, or beside the plane?

Figures 18 and 19 show the projected lines on the response surface to make it easier to see.

>Can the Authors propose a mechanism for why does the 16ply-P5 behaves >differently from other specimens?

Figure 1 and Figure 15 (a) shows tearing at the corners. As a fracture phenomenon of the corner part, surface shear fracture that occurs outside the cylinder shape is recognized. If the plate thickness is thick and the pitch is small, the influence between the corners may be considered in the surface shear failure.

Since this paper proposes a design technology for quasi-static performance, I would like to explain the quasi-static mechanism in the next paper.

>Is only 1 specimen used for every pitch and ply number? Can we be sure how >repeatable are the results?

There is one test piece each for this test. This test compares the number of corners installed and the thickness of the plate from the mechanism of surface shear failure. We believe that the reproducibility has been confirmed in the process of parameter study of the number of corners and plate thickness. The value of adjusted coefficient of determination( R’2 ) is also sufficiently high, indicating that it is stable. 

Again, thank you for giving us the opportunity to strengthen our manuscript with your valuable comments and queries.

Sincerely,

Tetsuya Gomi,

Reviewer 2 Report

The manuscript presents a very interesting and at the same time practical issue related to the analysis of an energy absorbing structure made of a material reinforced with carbon fiber. Despite its very practical nature, i have a few doubts about its scientific validity.

First of all, the introduction is incomplete. The introduction should introduce the reader to the issue, indicate the motivation to carry out the research, but also a critical review of the literature of the topic (see e.g. some works by Tomasz Gajewski on transverse shear in corrugated and sandwich panels) and detailed information on the new aspects contained in the manuscript. I suggest to supplement the introduction with the missing information.

Equation (1) - the lower bound of the integral should be 10 I suppose

line 209 - should be \beta_0

Apart from my minor comments, I fully support the publication of this manuscript in Applied Sciences, subject to some minor revisions.

Author Response

Dear Reviewer,

Thank you for providing these insights.
I think that your point made it possible to make the treatise easy to understand.
The response to comments is specified below. The corrections in the text are shown in red.

>Equation (1) - the lower bound of the integral should be 10 I suppose

There was a typographical error in the body.

Line 174,175 has been changed as follows.

The energy absorption (EA) is the total energy obtained from a load-displacement curve. As the evaluation zone of EA, we selected the displacement range between 0 mm and 60 mm where load had been generate in the experiment.

>line 209 - should be \beta_0

There was a typo in the text.

Fixed a typo.

Regarding the introduction, we added papers such as research on the static dynamic ratio of resin and crash boxes to the references.
This paper has shown that it proposes a design technique for quasi-static performance.(line90-line94)

Again, thank you for giving us the opportunity to strengthen our manuscript with your valuable comments and queries. 

Sincerely,

Tetsuya Gomi,

Round 2

Reviewer 1 Report

The manuscript has been significantly improved, and the Authors' explanations seem satisfactory.